# Identifying clinical features in primary care electronic health record studies: methods for codelist development

Jessica Watson,[1] Brian D Nicholson,[2] Willie Hamilton,[3] Sarah Price[3]

[1]Centre for Academic Primary Care, Bristol Medical School, University of Bristol, Bristol, UK
[2]Nuffield Department of Primary Care Health Sciences, University of Oxford, Oxford, UK
[3]University of Exeter Medical School, Exeter, UK

**Correspondence to**
Dr Jessica Watson;
jessica.watson@bristol.ac.uk

## ABSTRACT

**Objective** Analysis of routinely collected electronic health record (EHR) data from primary care is reliant on the creation of codelists to define clinical features of interest. To improve scientific rigour, transparency and replicability, we describe and demonstrate a standardised reproducible methodology for clinical codelist development.

**Design** We describe a three-stage process for developing clinical codelists. First, the clear definition a priori of the clinical feature of interest using reliable clinical resources. Second, development of a list of potential codes using statistical software to comprehensively search all available codes. Third, a modified Delphi process to reach consensus between primary care practitioners on the most relevant codes, including the generation of an 'uncertainty' variable to allow sensitivity analysis.

**Setting** These methods are illustrated by developing a codelist for shortness of breath in a primary care EHR sample, including modifiable syntax for commonly used statistical software.

**Participants** The codelist was used to estimate the frequency of shortness of breath in a cohort of 28 216 patients aged over 18 years who received an incident diagnosis of lung cancer between 1 January 2000 and 30 November 2016 in the Clinical Practice Research Datalink (CPRD).

**Results** Of 78 candidate codes, 29 were excluded as inappropriate. Complete agreement was reached for 44 (90%) of the remaining codes, with partial disagreement over 5 (10%). 13 091 episodes of shortness of breath were identified in the cohort of 28 216 patients. Sensitivity analysis demonstrates that codes with the greatest uncertainty tend to be rarely used in clinical practice.

**Conclusions** Although initially time consuming, using a rigorous and reproducible method for codelist generation 'future-proofs' findings and an auditable, modifiable syntax for codelist generation enables sharing and replication of EHR studies. Published codelists should be badged by quality and report the methods of codelist generation including: definitions and justifications associated with each codelist; the syntax or search method; the number of candidate codes identified; and the categorisation of codes after Delphi review.

## INTRODUCTION

Electronic health records (EHRs) have been used in routine primary care practice in the UK for at least 20 years.[1] EHRs are a rich resource for researchers and are increasingly used in epidemiological and medical research resulting in over 1500 publications since 2000, increasing from ~80 in 2005 to more than 450 in 2015/2016.

There are three well-established UK primary care EHR databases: the Clinical Practice Research Datalink (CPRD) including 4.4 million currently registered patients, covering 6.9% of the UK population[2]; The Health Improvement Network including 3.6 million currently registered patients giving ~5.7% coverage of the nation[3]; and QResearch including approximately 5 million currently registered patients in the UK.[4] All three databases record coded anonymised information about patients: demographics, diagnoses, symptoms, prescriptions, immunisation history, referral information and test results. Linkages enable follow-up of patients beyond the primary care setting, for example, to data recorded by the Office for National Statistics (ONS), the National Cancer Registration Service and to Hospital Episode Statistics. Integrated primary and

**Strengths and limitations of this study**

► This paper presents rigorous reproducible methods for codelist generation to increase transparency and replicability in electronic health record studies.
► Clear a priori definition of the feature of interest ensures clinical relevance and enables future researchers to assess the applicability of existing codelists to future research questions.
► Generation of auditable, replicable and modifiable syntax for codelists enables replication and 'future-proofs' codelists.
► Using a Delphi approach to reach consensus on inclusion of codes allows sensitivity analysis to explore the impact of uncertainty in coding.
► Using multiple clinicians in a Delphi panel reviewing codes may be unfeasible and inefficient for studies with large numbers of codes; a compromise of using two clinicians per feature from a panel of six offers a reasonable trade-off.

BMJ

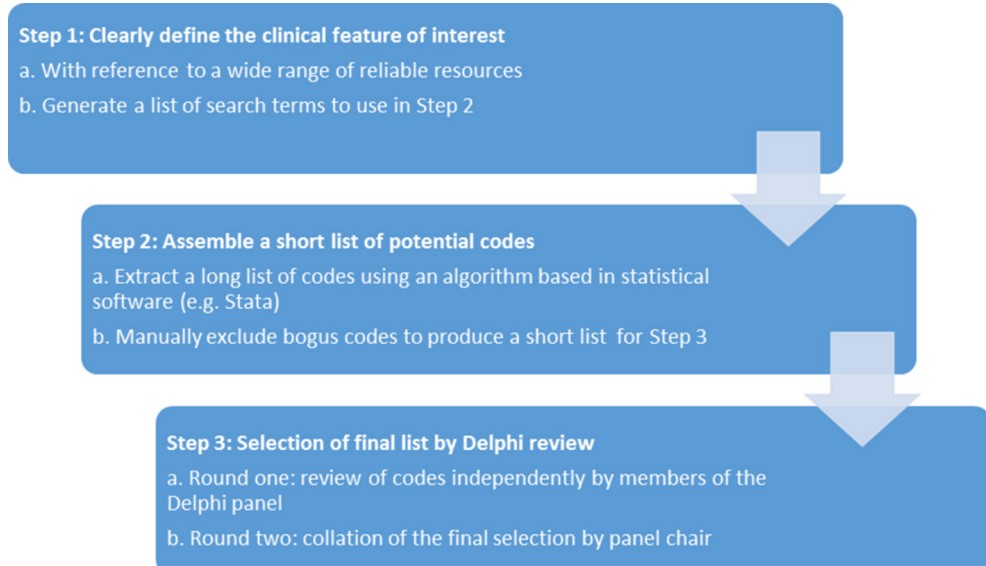

**Step 1: Clearly define the clinical feature of interest**

a. With reference to a wide range of reliable resources

b. Generate a list of search terms to use in Step 2

**Step 2: Assemble a short list of potential codes**

a. Extract a long list of codes using an algorithm based in statistical software (e.g. Stata)

b. Manually exclude bogus codes to produce a short list for Step 3

**Step 3: Selection of final list by Delphi review**

a. Round one: review of codes independently by members of the Delphi panel

b. Round two: collation of the final selection by panel chair

**Figure 1**  The method for codelist collation consists of three steps.

secondary care databases are also being developed. For example, ResearchOne includes data for over 5 million patients from General Practice, Child Health, Community Health, Out-of-Hours, Palliative Hospital, Accident and Emergency and Acute Hospital (http://www.researchone.org/).

A key stage in EHR research is identifying exposures and outcomes of interest. This apparently simple task is made more complicated by the fact that EHR clinical data is generally stored as codes, often including qualitative information, such as 'abdominal pain', 'left iliac fossa pain' and 'intermittent abdominal pain'. These separate codes need to be grouped into codelists or thesauri, with the groups containing all the codes pertaining to the variable of interest. However, the methods used to develop codelists are not standardised and are often poorly reported. They are an increasingly recognised source of bias in EHR research, owing to both inclusion of inappropriate codes and omission of important codes. To address this, the REporting of studies Conducted using Observational Routinely-collected health Data (RECORD) Statement states that '*a complete list of codes and algorithms used to classify exposures, outcomes, confounders, and effect modifiers should be provided*'.[5] Clinicalcodes.org has been developed by the University of Manchester to encourage researchers to publish clinical codelists used in EHR research,[6] and some other universities are developing their own open-access, citable repositories of codelists, for example, the University of Bristol[7] and University of Cambridge.[8] The current clinicalcodes.org repository contains 72 916 clinical codes deposited within 432 codelists (https://clinicalcodes.org), in the format of a list of papers and associated codes. This repository is a necessary step forward towards addressing transparency; however, it does not tackle the potential for bias,

as it is not sufficient to address the issues of scientific rigour and reproducibility in codelist development.

The problem is illustrated by brief examination of codelists recently deposited on the repository. Without a clear definition of the clinical variable a codelist is designed to encapsulate, it is not possible to critique or evaluate it for peer review or to decide whether it is generalisable to other studies. For example, codelists deposited for cancer (https://clinicalcodes.rss.mhs.man.ac.uk/medcodes/article/50/) do not adhere to the standardised International Classification of Diseases (ICD) definition of cancer, that is, ICD codes C00–C97, as 193 (~9%) of the 2254 Read codes related to carcinoma in situ (ICD D00–D09). Furthermore, 100 (~4%) codes were obsolete, or they indicated the absence of cancer or they were completely unrelated to cancer.

This demonstrates the need to establish standardised methods for codelist development. Currently, recommended methods, for example, Davé and Petersen[9] and CALIBERcodelists (http://caliberanalysis.r-forge.r-project.org/), need updating. This is because they omit steps to standardise the definition of clinical terms and because they are based in the Read code system, which is being superseded by SNOMED CT codes (Systematized Nomenclature of Medicine – Clinical Terms) in April 2018.

We have significant experience in EHR research, with ~40 published studies conducted in the CPRD since 2012. We have developed and refined rigorous methods for developing clinical codelists for use in CPRD studies independent of the Read code system. The aim of this paper is to report a clear, standardised, reproducible methodology and to increase scientific rigour in conduct of EHR research. The method is illustrated using the CPRD but applies equally well to other large EHR databases.

## METHODS

Our method for collating clinical codelists involves three stages, described in figure 1.

### Step 1: clearly define the clinical feature of interest (symptom, disease or illness) a priori

The first step is to clearly define the clinical feature of interest and establish inclusion and exclusion criteria. This requires clinical input, particularly from general practitioners (GPs) who are best placed to understand how clinical features are coded in a primary care setting. For rare conditions, which GPs encounter infrequently, it may also be important to get clinical input from hospital specialist doctors. Reliable sources of clinical information should be used, for example:

► International Classification of Primary Care (ICPC), which defines symptoms and diagnoses, provides synonyms for them and, importantly, lists what should be excluded from the definition.[10]
► The BMJ Best Practice guidelines (http://bestpractice.bmj.com/best-practice/welcome.html).
► National Institute for Health and Care Excellence (NICE) Clinical Knowledge Summaries (http://cks.nice.org.uk/).
► ICD-10 (http://apps.who.int/classifications/icd10/browse/2016/en) – this is less useful for symptoms, as it focuses on diseases.
► Medical Subject Headings (MeSH) (https://www.nlm.nih.gov/mesh/2016/mesh_browser/MBrowser.html).
► National Health Service (NHS) Digital Technology Reference data Update Distribution: https://isd.digital.nhs.uk/trud3/user/guest/group/0/home. Downloadable technology reference files including READ Code Browers with cross-map files.

Other potential resources include patient support groups, online discussion forums and already published codelists (eg, https://clinicalcodes.org). Hierarchical classifications such as Read, Systematized Nomenclature of Medicine (SNOMED) or ICD-10 may be useful for identifying additional search terms and synonyms.

For some symptoms, it is necessary to tailor the definition to the context of the disease under investigation. Abdominal pain is a good example, where pancreatic disease may cause pain in the epigastrium and left hypochondrium, whereas disorders in the sigmoid colon generate pain in the left iliac fossa.

### Step 2: assembling list of codes that may be used to record the clinical feature

The second stage consists of identifying all potential codes that might be used by GPs to record the clinical feature of interest defined in step 1 and collating them into a list.

This is done in several steps; we use Stata V.14 for this, but other software is possible.

---

**Box 1    Shortness of breath**

**International Classification of Primary Care (ICPC)**
► ICPC code: R02 (exclude: wheezing: R03; stridor: R04; hyperventilation: R98)

**BMJ Best Practice**
► Dyspnoea, also known as shortness of breath or breathlessness, is a subjective sensation of breathing discomfort (http://bestpractice.bmj.com/best-practice/monograph/862.html)

**National Institute for Health and Care Excellence Clinical Knowledge Summaries**
► Breathlessness is the distressing sensation of a deficit between the body's demand for breathing and the ability of the respiratory system to satisfy that demand. (http://cks.nice.org.uk/breathlessness#!backgroundsub)
► Breathlessness can be classified by its speed of onset as:
  – Acute breathlessness: when it develops over minutes, hours, or days.
  – Chronic breathlessness: when it develops over weeks or months.

**International Statistical Classification of Diseases, 10th revision (ICD-10)**
► ICD-10 code: R06: dyspnoea, orthopnoea, shortness of breath

**Medical Subject Headings (MeSH)**
► MeSH: difficult or laboured breathing. Breathlessness and dyspnoea.

**Patient forums**
► Puffed and winded.

**General practitioner colleagues**
► Consider including 'respiratory insufficiency'?

---

First, using the resources listed in step 1, an exhaustive list of synonyms for the outcome of interest is generated. Box 1 uses the example of shortness of breath.

Second, the lookup file of all medical codes provided by the CPRD (medical.txt)[i] is opened using Stata. This contains the alphanumeric Read code originally used by the GP to enter the clinical information, the CPRD's proprietary 'medcode' (which is simply a numeric equivalent of the Read code) as well as a verbal description (variable 'desc') common to both the medcode and Read code. A variable for the clinical outcome of interest (here, 'sob' for shortness of breath) is created and set to zero (see box 2). Then Stata searches the verbal description of each code, and sets 'sob' to 1 if it contains any of the synonyms. Example syntax to replicate this process in the statistical software package R is provided in the online supplementary material using the lookup file of all medical codes that come with the CPRD browsers. Note that, in this file, the verbal description is called 'readterm' rather than 'desc'.

The manual check for bogus codes should err on the side of caution, only rejecting codes that are clearly inappropriate according to predefined inclusion and exclusion criteria. Common reasons for exclusion are

---

[i]THIN and QResearch provide equivalent files.

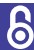

**Box 2  Syntax to search for potential codes using Stata**

```
insheet using "medical.txt", clear
*generate a binary variable for shortness of breath (sob) and set its
value to zero generate sob=0
/* search the verbal description of the Read code/medcode and
change the value of variable sob from 0 to 1 if it contains words that
suggest the code might be about the clinical feature of interest*/
replace sob=1 if regexm(desc, "[Ss]hortness [Oo]f [Bb]
reath|SHORTNESS OF BREATH")
replace sob=1 if regexm(desc, "[Ss][Oo][Bb]|SOB")
replace sob=1 if regexm(desc, "pnoea|PNOEA")
replace sob=1 if regexm(desc, "pnea|PNEA")
replace sob=1 if regexm(desc, "[Pp]uffed|PUFFED")
replace sob=1 if regexm(desc, "[Ss]hort [Oo]f [Bb]reath|SHORT OF
BREATH")
replace sob=1 if regexm(desc, "[Ss]hort|SHORT") & regexm(desc,
"[Bb]reath|BREATH") replace sob=1 if regexm(desc, "[Ww]
inded|WINDED")
replace sob=1 if regexm(desc, "[Dd]ifficult|DIFFICULT") &
regexm(desc, "[Bb]reath|BREATH") replace sob=1 if regexm(desc, "[Ll]
abour|LABOUR") & regexm(desc, "[Bb]reath|BREATH") replace sob=1 if
regexm(desc, "[Ll]abor|LABOR") & regexm(desc, "[Bb]reath|BREATH")
replace sob=1 if regexm(desc, "[Bb]reathless|BREATHLESS")
replace sob=1 if regexm(desc, "[Dd]istress|DISTRESS") &
regexm(desc, "[Bb]reath|BREATH") replace sob=1 if regexm(desc,
"[Dd]istress|DISTRESS") & regexm(desc, "[Rr]espir|RESPIR") replace
sob=1 if regexm(desc, "[Ii]suff|INSUFF") & regexm(desc, "[Bb]
reath|BREATH")
replace sob=1 if regexm(desc, "[Ii]suff|INSUFF") & regexm(desc, "[Rr]
espir|RESPIR")
/*order the dataset so that values of variable sob==1 are all placed
together*/
gsort sob
/* Manual check for bogus codes - manually change sob==1 to
sob==0 if the code is clearly inappropriate. */
edit medcode readcode desc sob
/*Retain only those codes that are specifically about sob*/ keep if
sob==1
/*Retain the variables of interest*/
keep medcode readcode desc sob
sort medcode
/*Save the file as a library for sob for the Delphi process*/
save "sob_library.dta", replace
/*Export as an Excel file
export excel using "sob_library", replace
```

that search terms can pick up bogus codes (eg, transobturator tape contains the letter sequence 'sob') or codes indicating a family history of a condition or screening for a condition rather than presence of a condition.

The output from step 2 is a list of potential codes that is then exported to Excel and reviewed manually in a Delphi-type process (step 3).

## STEP 3: DELPHI REVIEW OF CODES

The codelist is reviewed by one practising GP, plus at least one other GP from a panel of six, using a modified nominal group technique.[11] Each GP independently categorises the list, ranking each Read code/medcode using a three-point scale as follows:

1=definitely include: the code accurately defines the clinical feature of interest, and GPs would definitely use it.

2=uncertain: it remains unclear whether the code accurately reflects the clinical feature of interest, or whether GPs would use it.

3=definitely exclude: the code does not define the clinical feature of interest, and GPs definitely would not use it.

Panel members are encouraged to add comments explaining their reasons for exclusion or uncertainty, in the knowledge that these comments will be shared with an independent panel chair who will collate all of the results.

Codes are retained in the final list if they are ranked '1=definitely include' by at least one of the GPs, as this indicates sufficient evidence that the code may be used to record that clinical feature. Codes are dropped if they are ranked as '3=definitely exclude' or as '2=uncertain' by *all* reviewers.

An 'uncertainty' variable is also generated for retained codes to enable sensitivity analyses that remove codes for which any uncertainty exists about accuracy or use. The 'uncertainty' variable is defined as follows:

0='minimal uncertainty', as all panel members ranked the code as '1=definitely include'.

1='moderate uncertainty', at least one panel member ranked the code as '2=uncertain'.

2 = 'maximal uncertainty', at least one panel member ranked the code as '3=definitely exclude'.

Once the codelist has been generated, a frequency check may be performed using the study's dataset to identify the frequency of the clinical events attributed to each clinical code. If the Delphi process has been accurate, the most frequent events will most likely be coded as '0=minimal uncertainty', whereas there will be fewer events for the codes ranked as '1=moderate uncertainty' or as '2=maximal uncertainty'.

### Illustrative example using CPRD medical codes list

The library of codes for shortness of breath was used to estimate the frequency of this symptom in the year before diagnosis of lung cancer. Participants were CPRD patients aged over 18 years who received an incident diagnosis of lung cancer between 1 January 2000 and 30 November 2016.

Outcome measures included the number of patients reporting shortness of breath in the year before they were diagnosed with lung cancer, the proportion of all lung cancer patients reporting shortness of breath and the total number of episodes of shortness of breath.

In addition, a sensitivity analysis was carried out restricting the analysis to codes whose uncertainty variable was coded 0 (='minimal uncertainty'), that is, there was full agreement in the Delphi process that the code should be included.

**Table 1** Reasons for exclusion after first round of assessment

| Reason for exclusion | Number |
| --- | --- |
| Described 'apnoea'—absence of breathing—rather than breathlessness | 15 |
| Described negation of breathlessness | 2 |
| Described tachypnoea—abnormally rapid breathing—rather than breathlessness | 3 |
| Breathlessness related to pregnancy/neonate not pathology | 2 |
| Described hyperpnoea—increased rate and depth of breathing—not breathlessness | 1 |
| Description contained the string 'sob' but did not describe breathlessness (eg, removal of trans*ob*turator tape') | 6 |
| Total | 29 |

## RESULTS

The codelist generated for shortness of breath is presented here to illustrate the method we have described. The clinical resources reviewed in step 1 (see box 1 in Methods) indicated that the codes used to define shortness of breath should capture evidence of 'dyspn[o]ea', 'shortness of breath' (and its abbreviated term 'sob'), 'breathlessness', 'orthopn[o]ea', '"difficult" & "breathing"', '"labo[u]red" & "breathing"', '"breathing" & "discomfort"', 'puffed', 'winded', 'respiratory distress' and 'respiratory insufficiency'.

In step 2 (figure 1), Stata was used to produce a list of 78 possible shortness of breath codes (for syntax see box 2 in Methods). Of the 78 potential codes, 29 were excluded because they were clearly inappropriate (table 1).

The remaining codes were included in step 3: the Delphi review (online supplementary material table A1). Following the Delphi process, 49 codes were included in the final library (for complete list, see online supplementary material table A2). There was complete agreement to include 44 of the 49 (90%) of the codes, and partial disagreement over inclusion of just 5 (10%) of codes (figure 2). In this example, none of the codes were excluded during the Delphi process.

### Using codelists to identify symptoms

Of 28 216 patients diagnosed with lung cancer in the study, 7879 (28%) reported at least one episode of shortness of breath in the year before diagnosis. The total number of episodes of shortness of breath in the year before diagnosis was 13 091 (see table 2).

Of the 49 codes in the list for shortness of breath, 13 were never used by GPs to record this symptom (table 2). The majority of these were related to the BORG and CLASP breathlessness scores, and one was for respiratory insufficiency, highlighted as an uncertain code in the Delphi process.

Of the 37 codes used by GPs, 12 accounted for 90% of the total number of 13 091 episodes of shortness of breath

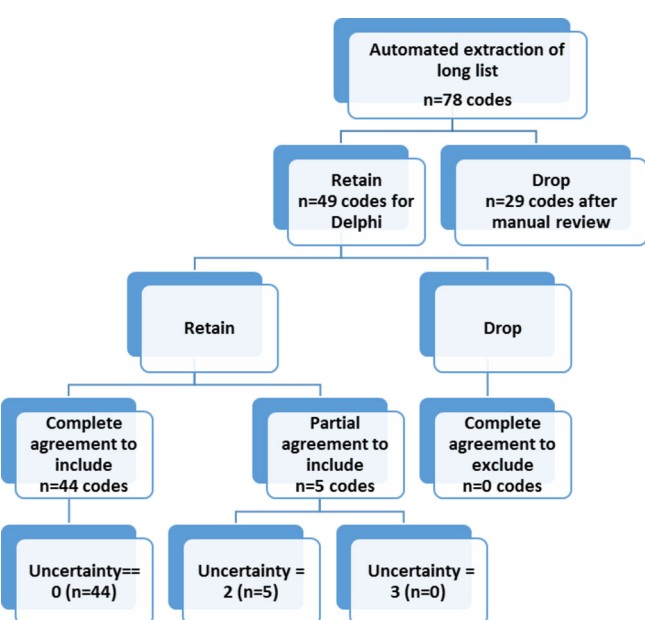

**Figure 2** Flow chart illustrating the selection of codes.

recorded. Furthermore, just four codes accounted for over 50% of the records (table 2).

### Sensitivity analysis

In the sensitivity analysis, the codelist was restricted to the 44 codes whose inclusion was fully agreed in the Delphi process. This resulted in the loss of just six patients reporting at least one episode of shortness of breath in the year before diagnosis (ie, the number fell from 7879 (28%) to 7873 (28%)). The total number of episodes of shortness of breath in the year before diagnosis was 13 081, compared with 13 091 using the complete codelist (see online supplementary material table A3, for complete list).

### DISCUSSION

We have presented a reproducible methodology for developing clinical codelists for use when conducting EHR research. It is intended to improve scientific rigour by standardising the conduct and reporting of this generally overlooked and under-reported stage of EHR research. These methods can be adapted to suit the needs of different EHR research questions. To facilitate this, we have included example syntax for two of the most widely used statistical software packages.

Reporting guidelines for observational studies aim to promote the core principles of the scientific process: discovery, transparency and replicability.[12] For systematic reviews, where searches for eligible papers are a core part of the methods, PRISMA guidelines stipulate that eligibility criteria, information sources used, search strategy and study selection process should be reported.[13] The process of searching for EHR codes is analogous to this. The RECORD statement requires '*a complete list of codes and algorithms*'; yet, what is meant by 'algorithms' is currently open to interpretation. We suggest that if EHR

**Table 2** Frequency of use of shortness of breath codes in the year before diagnosis with lung cancer

| medcode | Description | Frequency | Per cent | Cumulative % | Certainty variable* |
|---|---|---|---|---|---|
| 4822 | Shortness of breath | 3226 | 24.64 | 24.64 | 0 |
| 741 | [D]Shortness of breath | 1455 | 11.11 | 35.76 | 0 |
| 1429 | Breathlessness | 1116 | 8.52 | 44.28 | 0 |
| 19427 | MRC Breathlessness Scale: grade 2 | 1106 | 8.45 | 52.73 | 0 |
| 19426 | MRC Breathlessness Scale: grade 3 | 1010 | 7.72 | 60.45 | 0 |
| 5349 | Shortness of breath symptom | 816 | 6.23 | 66.68 | 0 |
| 5175 | Breathlessness symptom | 785 | 6.00 | 72.68 | 0 |
| 19430 | MRC Breathlessness Scale: grade 4 | 764 | 5.84 | 78.51 | 0 |
| 5896 | Dyspnoea – symptom | 437 | 3.34 | 81.85 | 0 |
| 2575 | Short of breath on exertion | 415 | 3.17 | 85.02 | 0 |
| 3092 | [D]Dyspnoea | 395 | 3.02 | 88.04 | 0 |
| 19432 | MRC Breathlessness Scale: grade 1 | 332 | 2.54 | 90.57 | 0 |
| 6326 | Breathless – moderate exertion | 261 | 1.99 | 92.57 | 0 |
| 19429 | MRC Breathlessness Scale: grade 5 | 189 | 1.44 | 94.01 | 0 |
| 2931 | Difficulty breathing | 187 | 1.43 | 95.44 | 0 |
| 12474 | SOBOE | 166 | 1.27 | 96.71 | 0 |
| 7932 | Breathless – mild exertion | 142 | 1.08 | 97.79 | 0 |
| 735 | [D]Breathlessness | 66 | 0.50 | 98.30 | 0 |
| 7000 | O/E – dyspnoea | 49 | 0.37 | 98.67 | 0 |
| 57903 | CLASP shortness of breath score | 44 | 0.34 | 99.01 | 0 |
| 31143 | Breathless – at rest | 39 | 0.30 | 99.30 | 0 |
| 7683 | Breathless – lying flat | 22 | 0.17 | 99.47 | 0 |
| 6434 | Paroxysmal nocturnal dyspnoea | 19 | 0.15 | 99.62 | 0 |
| 21801 | Breathlessness NOS | 10 | 0.08 | 99.69 | 0 |
| 11451 | [D]Orthopnoea | 9 | 0.07 | 99.76 | 0 |
| 9089 | Orthopnoea symptom | 8 | 0.06 | 99.82 | 0 |
| 24889 | Breathless – strenuous exertion | 5 | 0.04 | 99.86 | 0 |
| 7534 | O/E – respiratory distress | 4 | 0.03 | 99.89 | 1 |
| 18116 | Nocturnal dyspnoea | 3 | 0.02 | 99.92 | 0 |
| 2563 | Adult respiratory distress syndrome | 2 | 0.02 | 99.93 | 1 |
| 2737 | Dyspnoea on exertion | 2 | 0.02 | 99.95 | 1 |
| 24848 | Respiratory distress syndrome | 2 | 0.02 | 99.96 | 1 |
| 53771 | [D]Respiratory distress | 2 | 0.02 | 99.98 | 0 |
| 22094 | Borg Breathlessness Score: 10 maximal | 1 | 0.01 | 99.98 | 0 |
| 59860 | Borg Breathlessness Score: 4 somewhat. | 1 | 0.01 | 99.99 | 0 |
| 101843 | Short of breath dressing/undressing | 1 | 0.01 | 100.00 | 0 |
| 9297 | [D]Respiratory insufficiency | 0 | 0 | 100.00 | 1 |
| 37704 | O/E – orthopnoea | 0 | 0 | 100.00 | 0 |
| 42287 | Borg Breathlessness Score: 6 severe (+) | 0 | 0 | 100.00 | 0 |
| 57193 | Borg Breathlessness Score: 3 moderate | 0 | 0 | 100.00 | 0 |
| 57678 | Adult respiratory distress syndrome | 0 | 0 | 100.00 | 0 |
| 57759 | Borg Breathlessness Score: 2 slight | 0 | 0 | 100.00 | 0 |
| 60096 | CLASP shortness of breath score | 0 | 0 | 100.00 | 0 |
| 64049 | Borg Breathlessness Score: 5 severe | 0 | 0 | 100.00 | 0 |

**Table 2** Continued

| medcode | Description | Frequency | Per cent | Cumulative % | Certainty variable* |
|---|---|---|---|---|---|
| 67 566 | Borg Breathlessness Score: 9 very, very sev (almost maximal) | 0 | 0 | 100.00 | 0 |
| 68 707 | Borg Breathlessness Score: 1 very slight | 0 | 0 | 100.00 | 0 |
| 70 061 | Borg Breathlessness Score: 7 very severe | 0 | 0 | 100.00 | 0 |
| 70 818 | Borg Breathlessness Score: 0.5 very, very slight | 0 | 0 | 100.00 | 0 |
| 72 334 | Borg Breathlessness Score: 8 very severe (+) | 0 | 0 | 100.00 | 0 |
| Total | Total | 13 091 | 100.00 | 100.00 | |

*The 'certainty variable' is coded as: 0='minimal uncertainty' (all panel members agreed the code should be included in the list); 1='moderate uncertainty' (at least one panel member was uncertain that the code should be included); 2='Maximal Uncertainty' (at least one panel member thought the code should be excluded).
[D] terms are defined in the Read thesaurus as 'Symptoms, signs and ill-defined conditions'; CLASP, cardiovascular limitations and symptoms profile; MRC, medical research council; NOS, not otherwise specified; O/E, on examination; SOBOE, shortness of breath on exertion.

studies are to be transparent and reproducible, these algorithms should include: definitions associated with each codelist; the syntax or search method used; the number of candidate codes identified; and the categorisation of codes after Delphi review (see figure 2). This information could either be included within the published paper, as an appendix, or via online code repositories such as clinicalcode.org.

Benefits of this methodology include: the clear a priori definition of the clinical feature of interest based on reliable clinical resources; use of statistical software to comprehensively search all available codes; the iterative Delphi approach to reaching consensus on the most relevant codes; and the generation of an auditable, replicable and modifiable syntax for codelist generation enabling sharing and replication.

The way in which diagnosis is recorded in the EHR is heterogeneous, with different clinicians using different codes for the same clinical features. Definitions of clinical conditions also change over time, and codes are updated regularly in EHRs, often duplicating pre-existing codes. As a result, decisions about inclusion or exclusion of codes will vary between clinicians. Where Delphi panel members differ in decisions, free-text comments and discussions are important to understand these differences. In some cases, refinement of the a priori definition may be required to increase concordance between reviewers. However, residual differences are likely to persist owing to the inherent variability in clinicians' idiosyncratic patterns of coding. This variability can be captured by the sensitivity analysis using the 'uncertainty' variable to explore the impact of including or excluding these codes and by using the frequency check to identify which codes are used most often in the dataset.

Decisions about how to manage this uncertainty will depend on the research question and whether the aim is to increase sensitivity or specificity. In the example of breathlessness, we aimed to include *any code that might be used by a clinician to record this symptom,* in other words aiming to maximise sensitivity. Codes were therefore retained if either panel member ranked them as '*definitely include*', as this indicates that *some* clinicians may use this code to record this symptom.

Another option to enhance sensitivity when developing disease-specific codelists that has been described is the use of proxy codes. For example, one study included symptoms, referrals, tests or treatments indicative of the disease of interest, such as prescription of disease-modifying antirheumatic drugs as an indicator of rheumatoid arthritis. They found that 83.5% of 5843 patients had at least two indicator markers before a rheumatoid arthritis code was recorded.[14] This can be applied to symptoms, for example, using prescriptions of laxatives as a proxy for constipation in a study of colorectal cancer.[15]

For other research questions, it may be more important to focus on specificity, aiming to reduce the number of false-positive cases by using a narrower definition, with tighter inclusion and exclusion criteria. For these studies, it may be necessary to only include codes for which consensus exists to 'definitely include', and closer consensus may be reached among Delphi participants by increasing the number of Delphi rounds or the number of panel members. Criteria for inclusion of codes following Delphi review therefore depends on the purpose of the codelist. Researchers should make it clear whether codelists are sensitivity or specificity driven as this will affect the generalisability of the codelist to other studies.

Murphy *et al* suggested that a panel of at least six clinicians should be used for consensus methods.[11] This would be ideal to best capture the variability in coding between clinicians; however, this is unlikely to be an efficient use of clinicians' time for studies with large numbers of clinical codes, so a compromise of using two clinicians from a panel of six per clinical feature offers a reasonable trade-off. This is analogous to the methods for systematic reviews where two independent reviewers are routinely recommended. Using fewer than six GPs on the overall Delphi panel reduces the clinical styles incorporated and may not capture the inherent uncertainty in coding; it is therefore important that the extent of the clinical input

into the Delphi phase of the codelist review is clearly reported.

These challenges are demonstrated in the example provided; although both Delphi reviewers were 'certain' that MRC breathlessness 1 was a code indicating shortness of breath, further iterative feedback suggested that this actually indicates conditional breathlessness, being defined as '*not troubled by breathlessness except on strenuous exercise*'. This emphasises the importance of a transparent process of codelist development and illustrates the fact that this can be an iterative process, as Delphi reviewers, or later critics, may raise issues that require researchers to revisit and refine the definition or inclusion criteria to improve the sensitivity and specificity of the codelist.

### Comparison with existing literature

Previous studies have explored the implications of using differing code lists in EHR research. For acute stroke, significant differences were found between ONS codelists and a 'restricted' codelist developed by a Delphi panel, with very different mortality rates and different trends over time between these codelists.[16] Another study into coding of coronary heart disease in primary care found that limited code sets for 'angina' or 'myocardial infarction' unsurprisingly had limited sensitivity, with substantial proportions of coronary heart disease coded by non-specific codes.[17] Both these papers called for increased transparency and increased reporting of sensitivity analysis in EHR studies. Methods for compiling medical and drug code lists were presented by Davé and Petersen in 2009.[9] Their process was analogous to the second step described in our proposed methodology; however, it omitted the stage of defining clearly a priori the clinical feature of interest and the final stage of Delphi review, which is necessary to allow uncertainty to be explored using sensitivity analysis.

### Future implications

By April 2018, all primary care systems should have completed migration to an international clinical terminology called SNOMED CT (https://digital.nhs.uk/SNOMED-CT-implementation-in-primary-care), which does not share the same hierarchical structure as Read codes. This means that methods of codelist generation based on Readcodes can no longer be relied on.[9] Mapping SNOMED CT onto current coding systems is underway by the major EHR providers but will inevitably lead to a period of flux. By working independently of these hierarchical structures, using the description of the individual codes, we overcome these problems, allowing researchers to develop a search strategy that works across two or more classifications. Our proposed Delphi approach to code selection aims to reduce the impact of variable coding practice between clinicians. Clinicians are rarely trained in coding practice outside their individual clinical setting. An area of future development could therefore be for standardised coding training to be delivered as part of continued professional development.

## CONCLUSIONS

We suggest that as well as publishing codelists used in EHR studies, the methods used to generate these codelists should be reported. Collated codelists should be badged by quality according to whether they follow recommended methods for development.

As EHR research increases, it is important to avoid waste in research through incomplete or unusable research publications.[12] Although initially time consuming, using a rigorous and reproducible method for codelist generation 'future-proofs' the findings, and an auditable, modifiable syntax for codelist generation enables sharing and replication of EHR studies.

**Acknowledgements** We would like to thank Daniel Dedman, Senior Research CPRD, for his comments on an earlier draft of this paper, and Benjamin Feakins, statistician at the Nuffield Department of Primary Care Health Sciences at the University of Oxford, for writing the R syntax.

**Contributors** WH conceived and SP enhanced the methods of codelist collation described in the paper. JW wrote the original outline of the paper. SP designed and performed the data analysis. JW, SP and BDN developed the first draft of the paper. All authors contributed to subsequent drafts and read and approved the final manuscript.

**Funding** JW (DRF-2016-09-034) and BDN (DRF-2015-08-18) are both funded by Doctoral Research Fellowships from the National Institute for Health Research Trainees Coordinating Centre. WH is part-funded by the National Institute for Health Research (NIHR) Collaboration for Leadership in Applied Health Research and Care South West Peninsula at the Royal Devon and Exeter NHS Foundation Trust.

**Disclaimer** The views expressed are those of the authors and not necessarily those of the NHS, the NIHR or the Department of Health.

**Competing interests** None declared.

**Provenance and peer review** Not commissioned; externally peer reviewed.

**Data sharing statement** CPRD data on which the sensitivity analysis was based is held securely by University of Exeter Medical School under the CPRD data access licence (https://www.cprd.com/dataAccess/).

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
