## [Reviewer comments · BMJ Open]

ARTICLE DETAILS

TITLE (PROVISIONAL)	Identifying clinical features in primary care electronic health record studies: methods for codelist development
AUTHORS	Watson, Jessica; Nicholson, Brian; Hamilton, Willie; Price, Sarah

VERSION 1 – REVIEW

REVIEWER	Alex Dregan King's College London, UK
REVIEW RETURNED	27-Sep-2017

GENERAL COMMENTS	The authors are to be congratulated for a timely publication on the development of medical codes with primary care databases. This should be of great interest to an increasing number of researchers working with EHRs. I have only minor comments that the authors may wish to consider. Introduction The authors should provide similar figures for the three databases mentioned, eg both current and previous or only current. Methods I do not completely agree with the authors statement that GPs are best placed to understand the clinical features of diagnosis. Clinical experts have possible better insight into the recording of specific symptoms, as they represent the main source to start with. Page 9, first paragraph - the authors should also mention the updated NHS browser as an additional source to identify Read codes. Discussion Page 22 - Not sure why a panel of 6 clinicians is superior to a panel of , say 4 or 8? The authors may also wish to qualify in their Conclusion that, clinical diagnosis in routine care is heterogeneous (ie different clinicians use different words) and codes are updated regularly in EHRs, often duplicating pre-existing ones. The definition of a clinical condition also changes over time.
--

REVIEWER	Duncan Edwards University of Cambridge
REVIEW RETURNED	09-Oct-2017

GENERAL COMMENTS	This is a useful paper clearly describing a crucial element of electronic health record research methods, i.e. the development of codelists. Research using ehRs is increasing, so there is a need to improve the methods and transparency of codelist development. Relevant literature, databases, university websites and policies (e.g. plan to switch to SNOMED CT) are cited well, and the example is easy to follow. I have one minor criticism/suggestion: I am surprised that both of the Delphi reviewers were "certain" that MRC breathlessness 1 was a code indicating shortness of breath. My understanding is this indicates normality, and even MRC 2 could be normal for some patients (together these Read codes account for about 10% of the SOB cohort). I believe an annual MRC grading is requested to be recorded in COPD reviews as part of QOF (...so even with MRC3 and greater there is an argument to only include as an indicator of increased SOB in a patient if it is increased from their previous assessment). This questionable inclusion could be discussed in the conclusion and underlines rather than negates the point that codelists and their development should be transparent. Perhaps this could also be an example of how figure 1 can become iterative, as Delphi reviewers (or later critics) might raise issues/debates that require returning to step 1, at least in a limited way.
---

VERSION 1 – AUTHOR RESPONSE

2. Reviewer #1

Comment 2.1 The authors should provide similar figures for the three databases mentioned, eg both current and previous or only current.

Response: Thank you. We have amended so that all three databases quote current figures only.

Comment 2.2 I do not completely agree with the authors statement that GPs are best placed to understand the clinical features of diagnosis. Clinical experts have possible better insight into the recording of specific symptoms, as they represent the main source to start with.

Response: We agree that in some specific symptoms or specialist conditions hospital specialists have important insights and have amended the sentence:

This requires clinical input, particularly from GPs who are best placed to understand how clinical features are coded in a primary care setting. For rare conditions, which GPs encounter infrequently, it may also be important to get clinical input from hospital specialist doctors.

Comment 2.3 Page 9, first paragraph - the authors should also mention the updated NHS browser as an additional source to identify Read codes.

Response: Thank you we have added this as an additional bullet point:

- NHS Digital Technology Reference data Update Distribution (TRUD):

<https://isd.digital.nhs.uk/trud3/user/guest/group/0/home> Downloadable technology reference files including READ Code Browsers with cross map files.

Comment 2.4 Page 22 - Not sure why a panel of 6 clinicians is superior to a panel of say 4 or 8?

Response: The choice of 6 clinicians is based on the reference article. We have clarified this by adding the following sentence:

Murphy et al suggested that a panel of at least six clinicians should be used for consensus methods¹¹.

Comment 2.5 The authors may also wish to qualify in their Conclusion that, clinical diagnosis in routine care is heterogeneous (ie different clinicians use different words) and codes are updated regularly in EHRs, often duplicating pre-existing ones. The definition of a clinical condition also changes over time.

Response: Thank you, we have added the following sentence:

The way in which diagnosis is recorded in the EHR is heterogeneous, with different clinicians using different codes for the same clinical features. Definitions of clinical conditions also changes over time, and codes are updated regularly in EHRs, often duplicating pre-existing codes. As a result, decisions about inclusion or exclusion of codes will vary between clinicians.

We have also added a sentence in the conclusions:

Our proposed Delphi approach to code selection aims to reduce the impact of variable coding practice between clinicians. Clinicians are rarely trained in coding practice outside their individual clinical setting. An area of future development could therefore be for standardised coding training to be delivered as part of continued professional development.

3. Reviewer #2

Comment 3.1 I have one minor criticism/suggestion: I am surprised that both of the Delphi reviewers were "certain" that MRC breathlessness 1 was a code indicating shortness of breath. My understanding is this indicates normality, and even MRC 2 could be normal for some patients (together these Read codes account for about 10% of the SOB cohort). I believe an annual MRC grading is requested to be recorded in COPD reviews as part of QOF (...so even with MRC3 and greater there is an argument to only include as an indicator of increased SOB in a patient if it is increased from their previous assessment). This questionable inclusion could be discussed in the conclusion and underlines rather than negates the point that codelists and their development should be transparent. Perhaps this could also be an example of how figure 1 can become iterative, as Delphi reviewers (or later critics) might raise issues/debates that require returning to step 1, at least in a limited way.

Response: Thank you for this astute point. We accept the problem with this code, and the important lessons this raises about the fact that codelists may evolve in an iterative way. We have added the following sentences to the discussion:

These challenges are demonstrated in the example provided; although both Delphi reviewers were "certain" that MRC breathlessness 1 was a code indicating shortness of breath, further iterative feedback suggested that this actually defined as "not troubled by breathless except on strenuous exercise". This emphasises the importance of a transparent process of codelist development; and illustrates the fact that this can be an iterative process, as Delphi reviewers, or later critics, may raise issues which require researchers to revisit and refine the definition or inclusion criteria to improve the sensitivity and specificity of the codelist.

I hope this answers all the queries raised by the referees clearly. If you have any further questions please do let me know.

VERSION 2 – REVIEW

REVIEWER	Alex Dregan King's College London, UK
REVIEW RETURNED	21-Oct-2017
GENERAL COMMENTS	The authors addressed satisfactorily all previous concerns